# The T Cell Immunoscore as a Reference for Biomarker Development Utilizing Real-World Data from Patients with Advanced Malignancies Treated with Immune Checkpoint Inhibitors [note 1]

**DOI:** 10.3390/cancers15204913

**Published:** 2023-10-10

**Authors:** Islam Eljilany, Payman Ghasemi Saghand, James Chen, Aakrosh Ratan, Martin McCarter, John Carpten, Howard Colman, Alexandra P. Ikeguchi, Igor Puzanov, Susanne Arnold, Michelle Churchman, Patrick Hwu, Jose Conejo-Garcia, William S. Dalton, George J. Weiner, Issam M. El Naqa, Ahmad A. Tarhini

**Affiliations:** 1Departments of Cutaneous Oncology and Immunology, H. Lee Moffitt Cancer Center and Research Institute, Tampa, FL 33612, USA; 2Department of Machine Learning, H. Lee Moffitt Cancer Center and Research Institute, Tampa, FL 33612, USA; 3Department of Internal Medicine, Division of Medical Oncology, Comprehensive Cancer Center, The Ohio State University, Columbus, OH 43210, USA; 4Center for Public Health Genomics, School of Medicine, University of Virginia, Charlottesville, VA 22903, USA; 5Division of Surgical Oncology, Department of Surgery, School of Medicine, University of Colorado, Aurora, CO 80045, USA; 6USC Norris Comprehensive Cancer Center, Los Angeles, CA 90033, USA; 7Department of Neurosurgery, School of Medicine, University of Utah, Salt Lake City, UT 84132, USA; 8Huntsman Cancer Institute, Salt Lake City, UT 84132, USA; 9Oklahoma University Health Stephenson Cancer Center, Oklahoma City, OK 73104, USA; 10Department of Medicine, Roswell Park Comprehensive Cancer Center, Buffalo, NY 14263, USA; 11University of Kentucky Markey Cancer Center, Lexington, KY 40536, USA; 12Clinical & Life Sciences Department, Aster Insights, Hudson, FL 34667, USA; 13H. Lee Moffitt Cancer Center and Research Institute, Tampa, FL 33612, USA; 14Aster Insights, Hudson, FL 34667, USA; 15Department of Internal Medicine, Carver College of Medicine, University of Iowa Health Care, Iowa City, IA 52242, USA

**Keywords:** biomarker, immunoscore, immune checkpoint inhibitors, machine learning, oncology, real-world data, transcriptomics

## Abstract

**Simple Summary:**

This study utilized real-world data from patients with advanced malignancies who underwent immune checkpoint inhibitor (ICI) treatment. We used transcriptomic data to derive an immunoscore based on CD3+ and CD8+ T cell densities using CIBERSORTx and the LM22 gene signature matrix. The imputed immunoscore effectively predicted overall survival (OS); patients with an intermediate–high immunoscore achieved better OS than those with a low immunoscore. Therefore, the T cell immunoscore represents a promising signature for estimating OS with ICIs and can be used as a reference for future machine learning-based biomarker development.

**Abstract:**

Background: We aimed to determine the prognostic value of an immunoscore reflecting CD3+ and CD8+ T cell density estimated from real-world transcriptomic data of a patient cohort with advanced malignancies treated with immune checkpoint inhibitors (ICIs) in an effort to validate a reference for future machine learning-based biomarker development. Methods: Transcriptomic data was collected under the Total Cancer Care Protocol (NCT03977402) Avatar^®^ project. The real-world immunoscore for each patient was calculated based on the estimated densities of tumor CD3+ and CD8+ T cells utilizing CIBERSORTx and the LM22 gene signature matrix. Then, the immunoscore association with overall survival (OS) was estimated using Cox regression and analyzed using Kaplan–Meier curves. The OS predictions were assessed using Harrell’s concordance index (C-index). The Youden index was used to identify the optimal cut-off point. Statistical significance was assessed using the log-rank test. Results: Our study encompassed 522 patients with four cancer types. The median duration to death was 10.5 months for the 275 participants who encountered an event. For the entire cohort, the results demonstrated that transcriptomics-based immunoscore could significantly predict patients at risk of death (*p*-value < 0.001). Notably, patients with an intermediate–high immunoscore achieved better OS than those with a low immunoscore. In subgroup analysis, the prediction of OS was significant for melanoma and head and neck cancer patients but did not reach significance in the non-small cell lung cancer or renal cell carcinoma cohorts. Conclusions: Calculating CD3+ and CD8+ T cell immunoscore using real-world transcriptomic data represents a promising signature for estimating OS with ICIs and can be used as a reference for future machine learning-based biomarker development.

## 1. Introduction

Cancer is one of the world’s most complicated biological systems [1]. Notably, there are numerous conventional treatment modalities, including surgery, radiotherapy, molecularly targeted therapy, and chemotherapy, that have been developed [2]. Because of these advances, many patients can live longer and/or with a better quality of life. Other patients, however, do not respond to these treatments, so additional strategies are necessary [3]. One alternative approach is to induce an immune-mediated antitumor response. In this context, the discovery of immune checkpoints like cytotoxic T lymphocyte-associated antigen 4 (CTLA-4) and programmed death/ligand 1 (PD-1/PD-L1) has enabled the development of immune checkpoint inhibitors (ICIs), antibodies that target these checkpoints for the treatment of cancer [4]. Nevertheless, while ICIs can induce long-term remissions, even in patients with solid metastatic tumors, the majority of patients with other cancers do not achieve durable clinical responses [5,6,7]. One important reason for these low survival rates is the lack of prognostic biomarkers required to select patients for ICI monotherapies or combinations [8,9]. 

The development of ICIs clearly demonstrated that the immunogenicity of the tumor microenvironment (TME) plays an essential role in terms of the likelihood of response [10]. The TME can be infiltrated by various immune cells of potential immune-activating or suppressive effects with variable prognostic significance [11,12]. Indeed, several studies have indicated that helper T lymphocytes, cytotoxic T lymphocytes, and B lymphocytes play a crucial role as the prognostic markers of cancer patients [11,12,13,14,15,16,17,18]. The first evidence of tumors infiltrating lymphocytes (TILs) affecting survival was reported in 1921 [12]. Today, almost a century later, the prognostic value of TILs has become relevant to the staging of colon adenocarcinoma and with growing evidence to support prognostic value in melanoma [19]. 

The immunoscore, as validated by Galon et al. [20], estimates CD3+ and CD8+ T cell density in the core of the tumor (CT) and the invasive margin (IM). Evidence supports that a higher density of TILs (i.e., a high immunoscore) is associated with a better prognosis [11,21,22,23,24,25,26,27,28,29,30]. Hence, it is essential to note that immunoscore may reflect the immunogenicity of the TME and the tumor’s susceptibility to immunotherapy [31,32]. In fact, immunoscore has been widely evaluated in colorectal cancer patients to determine its effects on both overall survival (OS) and disease-free survival (DFS). For instance, a meta-analysis published recently found that a low immunoscore significantly correlated with poor OS (HR = 1.74, 95% CI: 1.43–2.13) and DFS (HR = 1.82, 95% CI: 1.64–2.03) [33]. Whereas a search of the literature revealed that few studies have investigated the role of immunoscore as a predictor for outcomes in other types of cancers, including esophageal cancer [34], bladder cancer [35], and non-small cell lung cancer (NSCLC), or in a cross-cancer capacity as a pan-cancer prognostic signature [36]. Furthermore, research into the intricate networks of cell–cell interactions in the TME and how this may affect several types of cancer responses to immunotherapies, such as ICIs, is a rapidly expanding area of research [36]. It relies on a limited repertoire of phenotypic markers and may result in cell loss or damage in immunohistochemistry or fluorescence-activated cell sorting [37]. To overcome these barriers and to take advantage of RNA sequencing data, system biology methods like transcriptome deconvolution were developed to estimate the relative densities of different cell types [36]. In this regard, evaluating TIL status in the tumor will be extremely helpful in investigating tumor immune cell states and susceptibility to immunotherapy. For this purpose, the specific objective of this study is to determine the prognostic value of immunoscore based on real-world transcriptomics data from TILs in patients with advanced malignancies treated with ICIs in an effort to explore its utility as a reference or control for machine learning-based biomarker development.

## 2. Methods and Materials

### 2.1. Patients and Datasets

For this study, real-world clinical and transcriptomic data retrospectively collected under the Total Cancer Care Protocol (NCT03977402) and Avatar^®^ project within the Oncology Research Information Exchange Network (ORIEN) of 18 collaborating cancer centers was utilized. An IRB-approved informed consent was obtained from all subjects at their participating institutions. The patients had to be 18 years old and above with cancer and have been treated with ICI. As shown in Figure 1, the construction process of the real-world immunoscore calculation is divided into different steps.

Briefly, this study involves the collection of normal and tumor tissue, blood and/or fluid samples with DNA in the form of frozen and/or formalin-fixed, paraffin-embedded (FFPE) tissue that was obtained prior to ICI treatment initiation from consenting subjects. Also, an additional tumor sample was obtained at the time of planned diagnostic biopsies or from previously collected or stored tumor tissue (if available). All patients’ related data, such as survey data, medical records data, cancer registry data, and other related data, were collected from the time they first joined the study until the time of conducting the current study.

### 2.2. RNA-Sequencing and Data Processing

RNA sequencing and data processing were completed as previously published https://www.asterinsights.com/white-paper/renal-cell-carcinoma-rwd-data/ (accessed on 28 September 2023). Subsequently, the RNA expression profiles were identified in the ORIEN database by downloading a series of matrix files that contain transcript per million (TPM) at the gene level. To avoid the value of zero during the normalization of TPM, a value of 1 was added to TPM, then Log10 transformation was used to normalize data value (TPM to (log_2_(TPM+1)); finally, the results were exponentiated to be a linear scale. Afterward, using linear regression, CIBERSORTx selected genes from the input matrix based on the LM22 signature matrix to deconvolve a given mixture. The input matrix of reference gene expression signatures was made using the standard annotation file. Finally, the CIBERSORTx algorithm runs in Python with 20 five-fold cross-validation permutations using the LM22 signature. The cut-off for statistical significance was set at *p* < 0.05. The input matrix of reference gene expression signatures was created using the standard annotation file.

### 2.3. Immunoscore Imputation

A limitation of our real-world dataset was that the direct calculation of the immunoscore was impractical, as our data did not include direct measurements of the CD3+ and CD8+ densities. However, it was shown that the gene expression levels can be used to impute cell type abundance. For this purpose, CIBERSORTx was employed using the leukocyte signature matrix (LM22), a gene signature matrix to impute the abundance of member cell types in the mixed cell population using gene expression data. Mainly, CIBERSORTx is an analytical tool that imputes gene expression profiles based on an input gene signature matrix. LM22 is a signature matrix containing 547 genes capable of accurately distinguishing 22 mature populations of human hematopoietic cells. This includes seven T cell types: naive and memory B, plasma, natural killer (NK), and myeloid subsets. Even though LM22 was made and tested with data from gene expression microarrays, it can also be used with data from RNA sequences [38]. Upon the imputation of CD3 and CD8 densities, patients’ immunoscore was calculated as categorical variables with two states of “low” and “intermediate and high”, following the procedure previously proposed by Galon et al. [1]. 

### 2.4. Construction of the Immunoscore

Following the percentile cut-off that is presented in Figure 2, the immunoscore was calculated as the average of the percentile of CD3_CT_ and CD8_CT_ densities of the patients among the training population. In order to avoid over-fitting, the immunoscore was computed using 20 iterations of five-fold validation such that in each iteration, four folds are employed to define the percentile functions, which are used to calculate the immunoscore of the patients in the remaining fold. Instantly, using this approach, the 20th percentile was calculated for each patient, with the ultimate immunoscore being computed based on their average.

Regarding the immunoscore categories, we considered two different cut-off points. The first cut-off point was the 25th percentile, as proposed by Galon et al. [1], and also shown in Appendix A. For the second cut-off point, we employed Youden’s J statistic to define the optimal cut-off point for our data. To do so, for each iteration of our cross-validations, a cut-off point was computed using Youden’s J statistic, with the final cut-off point being the average.

### 2.5. Statistical Analysis

Kaplan–Meier curves with 95% confidence interval (CI) survival plots as shaded areas were used to visualize the survival curves of patients with different immunoscores. In the survival analysis, the starting time point was the first exposure to ICI drugs, while death was considered an event and alive or lost follow-up was considered right-censored. A log-rank test was also employed to assess the significance of the difference in the survival distributions of the identified immunoscore categories. The analysis was performed first considering our entire dataset and second reflecting each cancer category individually. Statistical analyses were conducted using Python 3.8 libraries, Delaware, United States.

## 3. Results

### 3.1. Baseline Characteristics

Based on our methodology, data from 522 patients were included in our analysis. Of those patients, just over half (275 (53.0%)) experienced a study event, with 10.5 months as the median time to death. It is apparent in Table 1 that there is an imperceptible difference in the distribution of the four distinct malignant tumor types among our cohort, where each type represents just above or below a quarter of the total cohort, namely renal cell carcinoma (RCC), representing 28.5% of the overall cohort, followed by NSCLC (24.5%) and melanoma (23.9%), then 23% for head and neck cancer. In addition, data from this table indicates that the majority of the patients (80%) were treated with either nivolumab or pembrolizumab (42.0% and 38%, respectively), while a minority of patients (13.2%) were treated with the combination of ipilimumab and nivolumab. Moreover, only a smaller minority of participants (6.0%) received either ipilimumab, avelumab or cemiplimab. 

### 3.2. Entire Cohort

The first set of investigations examined the immunoscore analysis on our entire cohort using the 25th percentile cut-off. This identified 122 (23.4%) patients with a low immunoscore. However, a superior result was achieved when employing Youden’s J statistic, which identified the 43.5th percentile as the optimal cut-off point. This resulted in approximately doubling the number of patients (222 (42.5%)) with a low immunoscore. 

Figure 2A,B compare the Kaplan–Meier curves of the stratified patients based on the 25th and 43.5th percentile cut-off points, correspondingly. Both figures illustrate that those patients with an intermediate–high immunoscore (immunoscore = 1) achieved a significantly better survival time than patients with a low immunoscore (immunoscore = 0) (*p* < 0.001 for both cut-off points); however, based on Figure 2B, the 43.5th percentile resulted in a visually better stratification. Table 2 summarizes Harrel’s average c-index and the associated 95% confidence interval (CI). 

### 3.3. Cancer Categories

Next, we considered patients with the same histology as independent cohorts. We performed immunoscore sub-group analyses on the four cancer categories included in our dataset. The results obtained from the survival analysis and displayed in Kaplan–Meier curves of four different cancers histologies stratified using a 25th percentile cut-off point are presented in Figure 3. Remarkably, in Figure 3A,B, it is clear that patients with melanoma and head and neck tumors with an intermediate–high immunoscore (immunoscore = 1) had a significantly superior OS time than patients with a low immunoscore (immunoscore = 0) (*p* = 0.009 and *p* = 0.04, respectively). On the contrary, as shown in Figure 3C,D for OS time between an intermediate–high immunoscore (immunoscore = 1) and a low immunoscore (immunoscore = 0) in patients with NSCLC and RCC (*p* = 0.77 and *p* = 0.17, correspondingly), while we observe survival curve separation, the differences were not statistically significant.

Using Youden’s J statistic, different cut-off points were identified for the four cancer categories. Specifically, the optimal cut-off point of patients with melanoma was the 49.13th percentile; for head and neck cancer, it was the 43.7th; for NSCLC, it was the 57.75th; and for RCC, it was the 36.9th. Figure 4 presents the survival analyses of the different cancer histologies stratified based on the optimized cut-off points. Looking at Figure 4A,B, the statistically improved separation between the two immunoscore score categories for patients with melanoma and head and neck with log-rank test *p*-values of 0.01 and <0.001, respectively, are apparent. This could be interpreted as patients with melanoma or head and neck cancer, who achieved an intermediate–high immunoscore of 1, had a higher OS time than patients in the same cancer category who had a low immunoscore of 0. In contradiction to the above results, which indicated a significant separation of patients according to their immunoscore, Figure 4C,D demonstrated that adjusting the cut-off point of immunoscore in patients with NSCLC and RCC cancer did not result in a significant improvement in OS time between patients with an intermediate–high immunoscore versus patients with a low immunoscore (*p* = 0.37 and *p* = 0.25, respectively). Table 3 shows Harrel’s average C-index and the associated 95% CI together, and these results provide crucial insights into the role of immunoscore in predicting OS in various cancer types. 

## 4. Discussion

Although patients with select solid metastatic tumors have been shown to achieve long-term remissions with ICIs, most tumor types do not experience clinical benefits [5,6,7]. These low response rates may be partly attributed to the lack of prognostic biomarkers for identifying patients who have the capacity to benefit from therapy with ICIs [8,9]. The T cell immunoscore has been investigated as a prognostic biomarker that may also inform the status of an immune response to a tumor [30,31]. Due to the multiple parameters required for immunophenotyping, immunohistochemistry is limited in the number of immune groups that can be highlighted [39]. Also, flow cytometry measures a limited number of markers and requires strict technical methods, and therefore, closely related cell types may be missed [40].

In biomarker development with regard to ICI therapy, there is a need for a relatively simple biomarker signature of reasonable prognostic value that can be used as a reference, particularly in the rapidly advancing field of artificial intelligence and machine learning. This becomes more important when utilizing real-world data that are becoming increasingly available to researchers interested in ICI biomarker research. Clearly, bulk tissue gene expression profiling does not rely on surface markers or have cellular dissociation errors. Although immune-enriched gene expression signatures have prognostic value, connecting these signatures to specific TIL characteristics can be challenging [41]. Mathematically splitting bulk tumor gene expression patterns into individual cell types can inform this problem [20]. In addition to this, in reviewing the literature, little to no real-world data were found concerning the analysis of the association between immunoscore and ICIs outcomes in variant types of tumors, applying RNA expression data to calculate CD3+ and CD8+. Therefore, we attempted to validate the value of the CD3+ and CD8+ T cell-imputed immunoscore as a prognostic biomarker in cancer patients treated with ICIs and to support its utility as a “reference” or “control” in machine learning (ML)/artificial intelligence-based studies for biomarker development in the context of ICI and other immunotherapies when using transcriptomic data. So, a unique method was employed to accomplish this objective, utilizing transcriptomic data and CIBERSORTx, a computational approach that implements ML to determine imputed immunoscore and suggests immunoscore cut-off percentiles for different histologic cancer types [42].

One interesting finding is that immunoscore based on transcriptomics was able to significantly distinguish between the OS among patients with different cancer types as a total cohort using the cut-off point estimated by Galon et al. [1] or Youden’s J statistic, with better performance from the latter. The results showed that patients with intermediate–high immunoscore had better OS than patients with low immunoscore. A possible explanation for this might be that patients with a higher immunoscore have a higher density of TIL known to have antitumor activity [43]. Stratification was visually improved when Youden’s J statistic was applied. This result could cast a new light on the validity of our method of using transcriptomic data for analyzing immunoscore based on gene expression data from tumor tissue; CIBERSORTx might be a useful tool for determining the cellular composition of TIL. This includes both innate and adaptive compartments [42]. 

It is worth discussing the interesting facts revealed by the analysis results of each malignancy in our cohort. The analysis of immunoscore in patients with melanoma revealed a significant difference between patients with low versus intermediate–high immunoscore in both cut-offs: the 25th percentile and the 49.13th percentile cut-off point was estimated using Youden’s J statistic. This also aligns with previous reports that showed a significant association between higher CD8+, CD45+, and CD3+ cell counts and better OS (*p* = 0.001, *p* = 0.004, and *p* = 0.009, respectively) [44]. Even with these studies, it is becoming harder and harder to define immunoscore in melanoma based on pathological tumor-node-metastasis (TNM) staging because it is built on complex immune reactions intratumorally. Alternatively, metastatic lymph node tissue could be the best source for evaluating the immunoscore because, in many cases, the metastatic lymph nodes from a lymphectomy are the only available tissue [15]. A similar pattern of results was obtained by examining OS in patients with head and neck cancer based on their immunoscore with the two cut-off points used. The observed result is in agreement with the outcomes reported by Furgiuele et al. [45], who evaluated the prognostic role of TIL in developing an immunoscore in head and neck cancer patients. According to that study, the CD8+ density was an independent prognostic marker for recurrence-free survival (RFS) and OS. Furthermore, patients with high CD8+, CD68+, and FoxP3 T cell density had better OS. 

This study has not demonstrated a significant correlation between the generated T cell immunoscore and OS in patients with NSCLC or RCC when both cut-off percentiles were applied. The lack of significance in patients with NSCLC is contrary to that of X.-T. Li et al. (2021) [36], who found that patients with NSCLC could benefit from an immunoscore as a powerful, independent, and significant prognostic indicator. This could be related to the sample sizes in the different cohorts. Similarly, while our results in patients with RCC did not reach statistical significance, other studies support a significant correlation [46]. Researchers found that patients with a high immunoscore had prolonged disease-free survival (DFS), progression-free survival (PFS), and OS (HR 2.652, 2.848, and 2.933, respectively; all *p* < 0.001) compared to those with a lower score. These results were consistent in the sub-group analysis in patients with different Fuhrman grades and pathological TNM stages. 

A strength of the current research is that it is based on real-world data and investigated the prognostic value of immunoscore in multiple types of malignancies from transcriptomic data using ML for the first time. However, this study is not without limitations. Importantly, as RNA sequencing was performed on the specimens collected from the tumor, the immunoscore reflects the immune infiltration in a part of the tumor that could be central or peripheral, depending on the tumor biopsy obtained. This is unlike the original proposition of Galon, where the CD3+ and CD8+ T cell densities were best estimated via immunohistochemistry within the invasive margin [33,34,35,36]. In addition, there is no independent validation of the sample in this study because of its retrospective design. Future studies with larger cohorts are required to confirm the findings obtained in this study. 

## 5. Conclusions

In conclusion, the present research aimed to examine the prognostic value of immunoscore using real-world clinical and transcriptomics data in patients with various advanced malignancies treated with ICIs. The findings clearly indicate that the T cell immunoscore based on real-world transcriptomic data is able to predict patients at risk of death at a significant level and can be utilized as a reference for developing biomarkers based on machine learning.

## Figures and Tables

**Figure 1 cancers-15-04913-f001:**
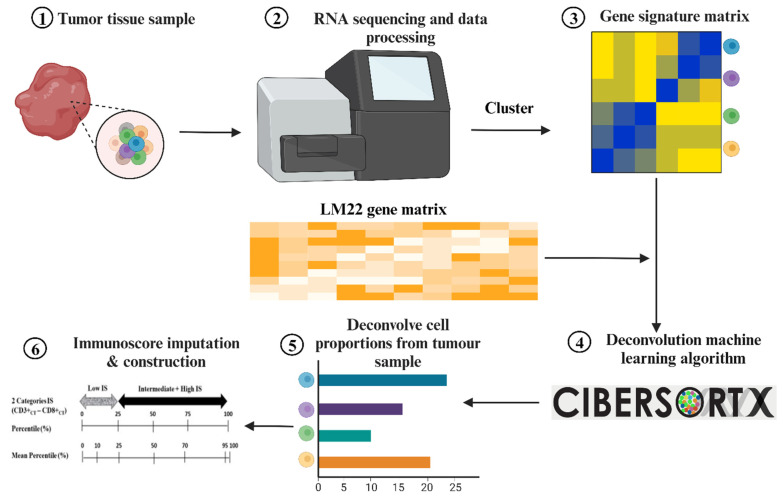
The construction process of the real-world immunoscore calculation. This figure demonstrates the different steps for calculating immunoscore based on real-world transcription data using CIBERSORTx as a learning tool.

**Figure 2 cancers-15-04913-f002:**
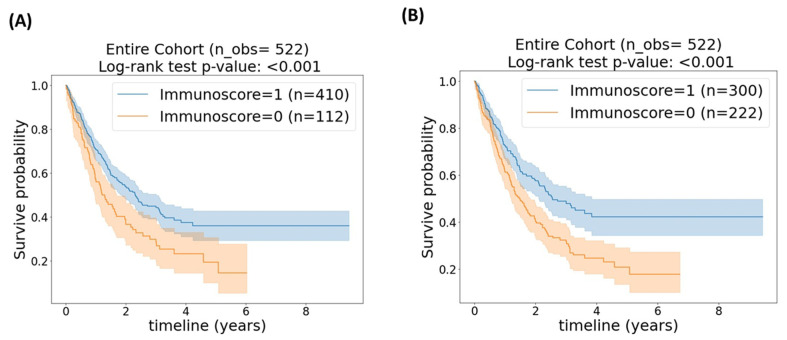
Kaplan–Meier plot of the entire cohort (*n* = 522) stratified based on (**A**) 25th percentile cut-off. (**B**) Kaplan–Meier plot of the entire cohort stratified based on (**A**) 43.5th percentile cut-off. *p*-value < 0.05 is considered statically significant.

**Figure 3 cancers-15-04913-f003:**
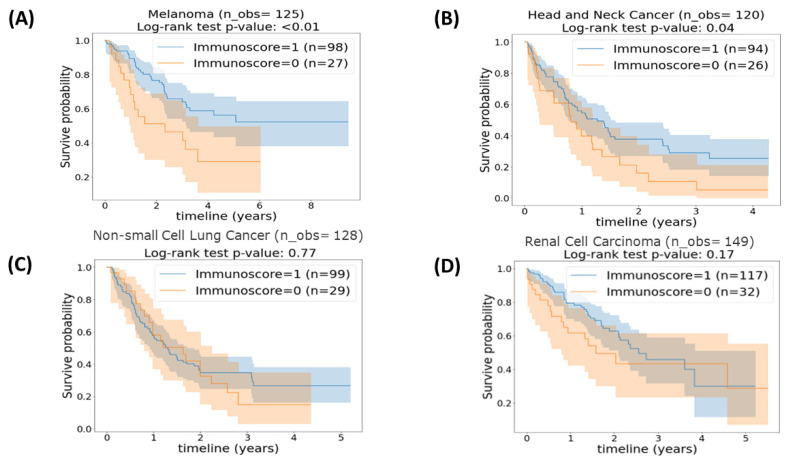
Kaplan–Meier plot of (**A**) patients with melanoma (*n* = 125) stratified based on the 25th percentile cut-off. (**B**) Patients with head and neck cancer (*n* = 120) stratified based on 25th percentile cut-offs. (**C**) Patients with non-small cell lung cancer (*n* = 128) stratified based on the 25th percentile cut-off. (**D**) Patients with renal cell carcinoma (*n* = 149) stratified based on the 25th percentile cut-off. *p*-value < 0.05 is considered statically significant.

**Figure 4 cancers-15-04913-f004:**
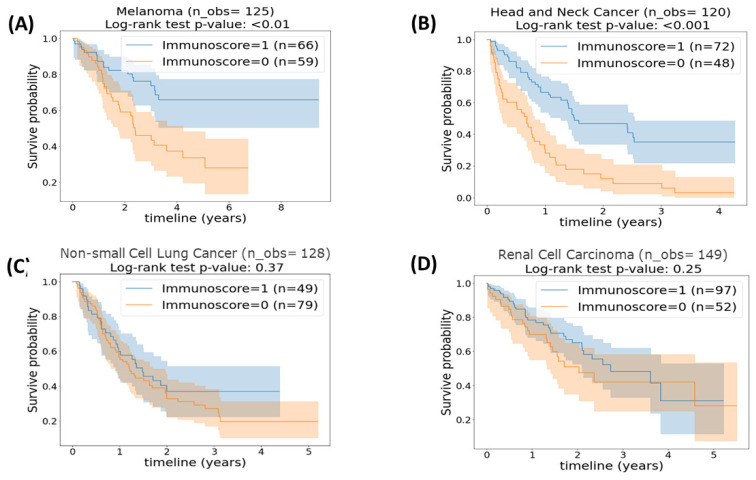
Kaplan–Meier plot of (**A**) patients with melanoma (*n* = 125) stratified based on a 49.13th percentile cut-off. (**B**) Patients with head and neck cancer (*n* = 120) stratified based on a 43.7th percentile cut-off. (**C**) Patients with non-small cell lung cancer (*n* = 128) stratified based on a 57.75th percentile cut-off. (**D**) Patients with renal cell carcinoma (*n* = 149) stratified based on a 36.9th percentile cut-off. *p*-value <0.05 is considered statically significant.

**Table 1 cancers-15-04913-t001:** Baseline disease and immunotherapy characteristics.

Variable	Patients (*N* = 522)
Age (in years)	
Median (range)	63 (19–90)
Sex, *n* (%)	360 (69)
Male	162 (31)
Female	
Race, *n* (%)	496 (95)
White	16 (3)
Black	10 (2)
Other	
ECOG performance status at diagnosis, *n* (%)	
0	99 (19)
1	84 (16)
2	10 (2)
Unknown	329 (63)
Cancer type, *n* (%)	
Renal cell carcinoma	149 (28.5%)
Non-small cell lung cancer	128 (24.5%)
Melanoma	125 (23.9%)
Head and neck cancer	120 (23.0%)
Prior systemic therapy, *n* (%)	
1 Prior line	198 (38)
2+ Prior line	324 (62)
First Immune checkpoint inhibitors, *n* (%)	
Nivolumab	219 (42.0%)
Pembrolizumab	202 (38.7%)
Ipilimumab + nivolumab	69 (13.2%)
Ipilimumab	30 (5.6%)
Avelumab	1 (0.2%)
Cemiplimab	1 (0.2%)

ECOG: Eastern Cooperative Oncology Group. 0: Fully active; no performance restrictions; 1: strenuous physical activity restricted; fully ambulatory and able to carry out light work; 2: capable of all self-care but unable to carry out any work activities, and up and about for >50% of waking hours; 3: capable of only limited self-care, and confined to bed or chair for >50% of waking hours; 4: completely disabled, cannot carry out any self-care, and totally confined to bed or chair.

**Table 2 cancers-15-04913-t002:** Average C-index and associated 95% Cl and log-rank *p*-values for the entire cohort.

Percentile Cut-Off	Avg. C-Index (95% CI)	Log-Rank Test *p*-Value *
Cut-off = 25th percentile	0.5402 (0.5345, 0.5459)	<0.001
Cut-off = 43.5th percentile	0.5528 (0.5466, 0.5591)	<0.001

* *p*-value < 0.05 was tested for significance using a log-rank test.

**Table 3 cancers-15-04913-t003:** Average C-index and associated 95% Cl and log-rank *p*-values per each cancer category.

Cancer Category	Percentile Cut-Off	Avg. C-Index (95% CI)	Log-Rank Test *p*-Value *
Head and neck	43.7th	0.55 (0.54, 0.56)	0.04
Renal Cell Carcinoma	36.9th	0.56 (0.55, 0.58)	0.17
Non-small cell lung	57.75th	0.47 (0.46, 0.48)	0.77
Melanoma	49.13th	0.58 (0.57, 0.59)	0.009

* *p*-value < 0.05 was tested using a log-rank test.

## Data Availability

The data sets used and analyzed for the current study are available from the corresponding author on request.

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
