# Peer review of "The T Cell Immunoscore as a Reference for Biomarker Development Utilizing Real-World Data from Patients with Advanced Malignancies Treated with Immune Checkpoint Inhibitorsâ€"

_cancers, 2023, doi:10.3390/cancers15204913_

Round 1
Reviewer 1 Report
The manuscript by Dr Eljilany and colleagues represents a logical progression of their earlier work in which they described the utility of CIBERSORT technology to objectively enumerate different hematopoetic cell types in tissue specimens, including tumor biopsies (references 38 and 42). The primary objective of the present study was to validate their transcriptomics/CIBERSORTx-based procedure as a method for enumeration of CD3+ and CD8+ T cells in tumor biopsies, which, in turn, could be applied to determine a prognostic immunoscore. This involved analysis of RNA sequence data derived from various types of solid malignancies, which, based on the differential expression of 547 genes, revealed 22 different cell types.
The study involved retrospective analysis of stored tumor tissue from 522 patients with cancers of four different types [melanoma (23.9%), head-and-neck cancer (23%), non-small cell lung cancer (NSCLC, 24.5%) and renal cell carcinoma (28.5%)]. All patients were treated with co-inhibitory immune checkpoint-targeted monoclonal antibodies, most prominently the PD-1 antagonists, nivolumab (42%) or pembrolizumab (38.7%). Using stringently calculated cut-off points, the immunoscore was categorized as either “low” or “intermediate/high” and correlated with treatment-related survival.
Briefly, with respect to analysis of data derived from the total cohort of patients, the authors observed a significant association of improved survival with higher immunoscores. However, notable differences emerged on subgroup analysis, with only melanoma and head-and-neck cancer showing significant associations of the immunoscore with a significant immunotherapy-related survival benefit. The authors concluded that “the T cell immunoscore based on real world transcriptomic data could significantly predict patients at risk of death and can be utilized as a reference for developing biomarkers based on machine learning”.
In my opinion the manuscript is well written and interesting, while the methods are technically sound. I do, however, have a few comments for the authors’ consideration:
1. What do you mean exactly by the term “real world” in the context of your study? Is it based on the number of patients, or the profile of different malignancies, or the number of different participating centers from which the patients were recruited, or the topical nature of checkpoint-targeted anti-cancer therapy, or all of these?
2. In addition to checkpoint inhibitors, did the participating patients receive any other types of anti-cancer therapy?
3. Using CIBERSORTx-based technology, what is the projected cost of the immunoscore test?
4. Do you intend undertaking a comparison between your method of determining the immunoscore and the Galon immunochemistry/digital pathology method?
5. If in your possession, can you include additional, presumably relatively accessible, comparative data from relatively undemanding biomarker investigations such as tumor PD-L1 expression, levels of systemic soluble PD-L1, circulating lymphocyte and platelet counts, NLR and C-reactive protein?
Author Response
To: Reviewer 1,
Journal of Cancers
Resubmission Date: September 13th, 2023
Dear Valued Reviewer,
Thank you for considering our manuscript “The T Cell Immunoscore as A Reference for Biomarker Development Utilizing Real-world Data from Patients with Advanced Malignancies Treated with Immune Checkpoint Inhibitors’ ID (cancers- 2601811) for review. We are grateful for the time and effort you have dedicated to providing valuable feedback on our manuscript. We have been able to incorporate changes that reflect your suggestions and have highlighted the changes within the manuscript in track changes.
Comment 1: What do you mean exactly by the term “real world” in the context of your study? Is it based on the number of patients, or the profile of different malignancies, or the number of different participating centers from which the patients were recruited, or the topical nature of checkpoint-targeted anti-cancer therapy, or all of these?
Response 1: "Real-world data" in the context of our study refers to the fact that our data is derived from the electronic health records (EHR) of consenting subjects receiving standard-of-care treatment within 18 collaborating cancer centers who had consented to have their data used for research. The data was collected retrospectively and was not part of a clinical trial.
Comment 2: In addition to checkpoint inhibitors, did the participating patients receive any other types of anti-cancer therapy?
Response 2: No.
Comment 3: Using CIBERSORTx-based technology, what is the projected cost of the immunoscore test?
Response 3: CIBERSORTx is a free online tool. The main cost for our studies was related to the RNA sequencing procedure as part of the AVATAR project of the Total Cancer Care protocol to which study subjects had consented.
Comment 4: Do you intend to compare your method of determining the immunoscore and the Galon immunochemistry/digital pathology method?
Response 4: Thank you for your suggestion; this can be a future direction for our research, although limited by the availability of tumor tissue at this time.
Comment 5: If in your possession, can you include additional, presumably relatively accessible, comparative data from relatively undemanding biomarker investigations such as tumor PD-L1 expression, levels of systemic soluble PD-L1, circulating lymphocyte and platelet counts, NLR and C-reactive protein?
Response 5: Thank you for your suggestion. Unfortunately, this data is not available to us currently.

Reviewer 2 Report
This study is interesting with clinical significance. Immunotherapy therapy has revolutionized the tumor therapy. The authors put forward a new and comprehensive point of view on machine learning-based biomarker for determining the prognostic value. The followings are some comments to the authors.
Comments:
1.I suggesting providing baseline information of 522 patients in this study, including age, sex (male or female) , ECOG performance status, number of metastatic organs, extent of disease (metastatic or locally advanced), prior systemic therapy (1 prior line, 2 prior lines, 3 prior lines). because the factors above play an important role in OS.
2.The sample size of immunoscore-1 and immunoscore-0 is very different. How to explain the influence of sample size imbalance between groups on the results?
Author Response
To: Reviewer 2,
Journal of Cancers
Resubmission Date: September 13th, 2023
Dear Valued Reviewer,
Thank you for considering our manuscript “The T Cell Immunoscore as A Reference for Biomarker Development Utilizing Real-world Data from Patients with Advanced Malignancies Treated with Immune Checkpoint Inhibitors’ ID (cancers- 2601811) for review. We are grateful for the time and effort you have dedicated to providing valuable feedback on our manuscript. We have been able to incorporate changes to reflect your suggestions and have highlighted the changes within the manuscript in track changes.
Here is a point-by-point response to your comments and concerns.
Comment 1: I suggest providing baseline information of 522 patients in this study, including age, sex (male or female), ECOG performance status, number of metastatic organs, the extent of disease (metastatic or locally advanced), prior systemic therapy (1 prior line, 2 prior lines, 3 prior lines). Because the factors above play an important role in OS.
Response 1: Thank you for your suggestion. We have added this information with available data to Table 1.
Comment 2: The sample size of Immunocore-1 and Immunocore-0 is very different. How to explain the influence of sample size imbalance between groups on the results?
Response 2: We appreciate the reviewer's comment.
In our analysis, we anticipated a higher rate of censoring events in the Immunocore-0 group compared to the Immunocore-1 group, necessitating a larger sample size in the Immunocore-1 group to ensure an adequate number of events for analysis. To further validate our results and mitigate the potential influence of imbalance in sample size, we conducted subcategory analyses, where we stratified by cancer type and changed the cut-off point to confirm the robustness of our findings.

Round 2
Reviewer 1 Report
The authors, where possible, have made a satisfactory attempt to address my comments and I recommend that the revised manuscript be accepted for publication.